# Discovery of Natural Small Molecules Promoting Collagen Secretion by High-Throughput Screening in *Caenorhabditis elegans*

**DOI:** 10.3390/molecules27238361

**Published:** 2022-11-30

**Authors:** Jinyang Fang, Xinyue Wu, Xi’nan Meng, Dejin Xun, Suhong Xu, Yi Wang

**Affiliations:** 1Pharmaceutical Informatics Institute, College of Pharmaceutical Sciences, Zhejiang University, Hangzhou 310058, China; 2Department of Burn and Wound Repair of the Second Affiliated Hospital, Zhejiang University School of Medicine, Hangzhou 310058, China; 3International Biomedicine-X Research Center of the Second Affiliated Hospital, Zhejiang University School of Medicine, Zhejiang University—University of Edinburgh Institute, Haining 314400, China; 4Center for Stem Cell and Regenerative Medicine, Zhejiang University School of Medicine, Hangzhou 310058, China; 5Innovation Institute for Artificial Intelligence in Medicine, Zhejiang University, Hangzhou 310018, China

**Keywords:** high throughput drug screening, collagen synthesis and secretion, natural small molecule, COL-12, *C. elegans*, transfer learning

## Abstract

Advancing approaches for drug screening are in great demand to explore natural small molecules that may play important roles in collagen biogenesis, secretion, and assembly, which may find novel lead compounds for treating collagen-related diseases or preventing skin aging. In this study, we generated a single copy insertion transgenic P*col-19*- COL-12::GFP *Caenorhabditis elegans* (*C. elegans*) strain to label epidermis collagen XII (COL-12), a cuticle structure component, and established an efficient high-content screening techniques to discover bioactive natural products in this worm strain through quantification of fluorescence imaging. We performed a preliminary screening of 614 compounds from the laboratory’s library of natural small molecule compounds on the COL-12 labeling worm model, which was tested once at a single concentration of 100 µM to screen for compounds that promoted COL-12 protein amount. Besides *col-12*, the transcriptional levels of worm-associated collagen coding genes *col-19* and sqt-3 were also examined, and none of the compounds affected their transcriptional levels. Meanwhile, the protein levels of COL-12 were significantly upregulated after treating with Danshensu, Lawsone, and Sanguinarine. The effects of these drugs on COL-12 overexpressing worms occur mainly after collagen transcription. Through various validation methods, Danshensu, Lawsone, and Sanguinarine were more effective in promoting the synthesis or secretion of COL-12.

## 1. Introduction

Collagen is a biopolymer found in the animal body, and one of the most abundant proteins in the extracellular matrix that provides strength to the skin, joints, and bones in the human body [1]. It plays an important role in anti-aging, scar formation, and other biological mechanisms. There are 28 known collagen isoforms (collagen I to XXVII), which vary in structure and function [2]. The molecular conformation of the collagen triple helix confers strict amino acid sequence constraints, requiring a (Gly-X-Y)(n) repeating pattern and a high content of amino acids. The increasing family of collagens and proteins with collagenous domains shows the collagen triple helix to be a basic motif adaptable to a range of proteins and functions [3]. Abnormal collagen biogenesis leads to dysplasia in bone or skin, such as Schmid metaphyseal chondrodysplasia [4] and Ehlers–Danlos syndrome [5]. Even excessive fibrosis associated with wound healing and skin aging is related to the disorder of collagen biogenesis [6]. Though the structures and functions of collagen molecules have been gradually explored, more information on the regulation of collagen biogenesis remains inadequate.

In the human body, collagen VI may play a role in special tissues related to tensile stress and in maintaining the functional integrity of skeletal muscle [7]. Human α5 was found to be localized in the outer epidermis and COL6A5 has been associated with atopic dermatitis and designated COL29A1. Atopic dermatitis patients were found to lack collagen VIα5 expression in the outer epidermis [8]. However, the mechanisms of collagen VI biogenesis are still unclear. 

In order to reduce the difficulty of research, the basic strategy of research in the field is to explore molecular mechanisms in simple organisms and verify the conservatism of these mechanisms in higher mammals. *Caenorhabditis elegans* (*C. elegans*) is a tiny, free-living nematode with a simple feeding procedure, fast reproduction, and a transparent body suitable for microscopic observation and genetic screening [9]. *C. elegans* is generally used in genetics, biophysics, immunology, neuroscience, and other related research fields [10]. This nematode is enclosed within a unique extracellular matrix called the cuticle [11]. The cuticle’s major constituent is collagen, synthesized and secreted apically from epithelial cells and polymerized on the external epithelial surface [12]. Therefore, we selected *C. elegans* as the model for further screening.

Collagen XII (COL-12) is homologous to the COLⅥA5 of human and a structural component of the cuticle and a part of the collagen trimer. To monitor and quantify this collagen protein, we generated a single copy insertion transgenic *C. elegans* strain which explicitly expresses the GFP fused COL-12 in the epidermis, referred to as the COL-12 worm model [13]. The high-throughput drug screening technology demands small samples, and is high automated, rapid and sensitive [14]. With the help of high-throughput drug screening technology we can monitor the protein mass of COL-12 through fluorescent intensity. It can also be used to study collagen secretion and assembly. In recent years, small molecule compounds have been the choice for new drug development, providing an important basis for treating many diseases. With the continuous optimization of high-throughput and high-content screening methods, large-scale screening of small-molecule compound libraries has become feasible and effective [15]. Our work combines the characteristics of *C. elegans* and the optimization of high-content imaging conditions to conduct high-throughput screening of compounds promoting collagen secretion on this model, and obtains several compounds (Danshensu, Lawsone, and Sanguinarine) that may promote collagen synthesis and secretion, and preliminarily explores the regulation mechanisms.

## 2. Results and Discussion

### 2.1. COL-12 Single Copy Insertion Transgenic Strain Construction in C. elegans

The single copy insertion of the gene of interest expression has been known as a stable native gene expression method in *C. elegans* [16]. Template-dependent homologous recombination can generate the gene of interest at chromosomal sites. To quantify COL-12 protein expression in C. elegans, we established a method by inducing CRISPR/cas9 system and self-excising cassette to generate a single-copy transgenic worm strain at specific loci. The repair template contained *col-19* promotor driven GFP fused COL-12 cDNA sequence flanked by two DNA homologous arms, which are homologous to the genomic regions at the loci ttTi4348 on Chromosome I. Plus, a self-excising cassette composed of hygromycin B resistance and P*hsp-16.41* driven Cre sequence between two loxP sites in the same direction was integrated into the template for convenient transgenic worm screening (Figure 1).

To examine the epidermal-specifically expressed COL-12 of this transgenic worm, we imaged the worms by spinning disk confocal microscope. The Day 1 adult (D1) worms showed green signals on the cuticle (Figure 1). This indicates that the single copy insertion allele expressed COL-12 in the transgenic strain is expressed and localized on cuticle.

### 2.2. Development of High-Content Screening Approach for Prompting COL-12 Expression

To select the appropriate imaging conditions for the COL-12 overexpression worm model for screening, four different imaging platforms were used: Microscope (Leica, Wetzlar, Germany), Confocal (Andor, UK), High-Throughput Confocal (Molecular Devices, San Jose, CA, USA), and High-Throughput PICO (Molecular Devices, San Jose, CA, USA); and we used different plate types: plate1 (96-well plate black, 3904, Corning, Corning, NY, USA), plate2 (96-well plate black, 655090, Grenier, Frickenhausen, Germany), plate3 (96-well plate transparent, 701001, Nest Biotechnology, Woodbridge, NJ, USA). The final images were acquired using each instrument at different plate types with different magnification objectives (Figure 2). According to the classification of the screening method combined with the problems in the actual imaging process, we divided the shots into several groups, which are: (1) Microscope + Corning, 3904 + 5× lens; (2) Microscope + Corning, 3904 + 10× lens; (3) Microscope + Grenier, 655,090 + 5× lens; (4) Microscope + Grenier, 655,090 + 10× lens; (5) Microscope + Nest, 701,001 + 5× lens; (6) Microscope + Nest, 701,001 + 10× lens; (7) High-Throughput PICO + Corning, 3904 + 4× lens; (8) High-Throughput PICO + Grenier, 655,090 + 4× lens; (9) High-Throughput Confocal + Corning, 3904 + 4× lens; (10) High-Throughput Confocal + Grenier, 655,090 + 4× lens; (11) Confocal + Corning, 3904 + 4× lens; (12) Confocal + Grenier, 655,090 + 4× lens.

To optimize the imaging conditions by reducing the effects of neighbor hole apertures, the Nest 701001 plate type was excluded. The instruments Microscope and Confocal were excluded due to manually photographing to obtain high throughput screening efficiency. To achieve sufficient worm density for subsequent single worm separation, a 10× lens was excluded. From the data analysis related indicators, the images obtained from PICO are more suitable for cutting by Scellseg software, the nematode images obtained are complete, so the overall area is large, and secondly, the plate fluorescence intensity related indicators such as Mean Gray Value and Integrated Density of plate2 are more stable than plate1. These requirements determine (8) High-Throughput PICO + Grenier, 655,090 + 4× lens is the best combination for the image sampling conditions in the screening assay (Figure 2D).

### 2.3. Screening for Active Ingredients That Promote COL-12 Expression

After imaging, the images were imported into the software Scellseg (Pharmaceutical Informatics Institute, College of Pharmaceutical Sciences, Zhejiang University, Hangzhou, China) [17] to label the images for training manually, the training set was established, and the images were cropped to obtain single worm images (Figure 3A). Finally, the single worm images were processed using ImageJ to obtain fluorescence intensity data. Meanwhile, for the initial screening, to identify the feasibility of single nematode isolation and the fluctuation of each nematode data in the same picture, we analyzed the Integrated Density, Mean Gray Value and Area of Control group and Vitamin A (it has been shown to have a stimulating effect on collagen production), and concluded that the application of Scellseg software is feasible and the nematode screening method is more stable (Figure 3B,C).

We screened 614 compounds by the method (Figure 4A), expecting to find compounds that promote the COL-12 amount. The preliminary screening results show the Mean Gray Value for individual nematodes in the effective compound group from the screening results (Figure 4B). These results indicate that 26 compounds enhance nematode epidermal fluorescence intensity (Figure 4C).

### 2.4. Validation Study of the High Magnification Imaging Effect of Compounds Promoting COL-12 Expression

The safe concentration of the drug was used as our dosing concentration and co-cultured with worms at 20 °C for 12 h. Afterward, the worms were photographed by spinning disk confocal. Six compounds with significant effects were screened for Danshensu (Yuanye, Shanghai, China), Lawsone (DESITE, Chengdu, China), Quercetin (Yuanye, Shanghai, China), Hematoxylin (DESITE, Chengdu, China), Shikonin (DESITE, Chengdu, China) and Sanguinarine (Yuanye, Shanghai, China) in promoting collagen amount (Figure 5A). The fluorescence intensity of the COL-12 nematode model was then observed and analyzed, and the mean grey scale values and intensity density were calculated (Figure 4C and Figure 5B). The effect of different small molecular compounds on collagen distribution forms is evident from the images, but further experiments are still needed to confirm the forms of action of the drugs.

### 2.5. Validation of the Role of Natural Compounds

To further verify whether the compounds affect *col-12* transcription level and to preliminarily investigate the sites of action, we tested the levels of worm-associated collagen transcription and protein expression after being co-cultured with the drugs. The *col-19*, *col-12*, and *sqt-3* transcriptional levels were examined in N2, and we found that the transcriptional level of *col-12* was induced by Danshensu, Quercetin and Shikonin exhibiting a rising trend. It significantly increased after Hematoxylin-induced, and no significant change after Lawsone and Sanguinarine action was found. (Figure 6A) The transcription level of *col-19* treated by Danshensu, Lawsone, Shikonin and Sanguinarine exhibited a rising trend, It significantly increased after Hematoxylin-induced, and decreased after Quercetin action. (Figure 6B) The transcription level of *sqt-3* was induced by Danshensu, Lawsone, Quercetin, Hematoxylin and Sanguinarine slightly, and decreased after Shikonin action (Figure 6C). Meanwhile, COL-12 protein levels were significantly upregulated after the treatment of Danshensu, Lawsone, and Sanguinarine (Figure 6D,E). The above results suggest that the effects of these drugs on COL-12 overexpressing worms occur mostly after collagen transcription. Plus, the results showed that Lawsone was more effective in promoting the COL-12 biogenesis in epidermis. Since Lawsone was toxic to nematodes at a concentration of 1000 μM, we decided to use 100 μM Lawsone as the working concentration (Figure 6F). The drug had a clear dose-effect relationship as verified by doubled and half dilution (Figure 6G).

## 3. Materials and Methods

### 3.1. COL-12 Labelin g Worm Model Construction

Plasmid pSX2796 (pCFJ210-LoxP-SEC-Loxp-P*col-19*-COL-12::GFP) was constructed. pCFJ210 vector was used as the backbone for generating plasmid, and primers were designed according to the insertion site and the coden sequence of *col-12* (the sequences of which were ggacccttggagggtacaggATGACCGAAGATCCAAAGCAGATTGCCC and CCTCCCGAACCTCCACCTCCATATCCTGGAGCGGTACGTGGTGGTGGGC). P*col-19* promoter was used for an epidermal-specific expression. The ttTi4348 genome locus was selected for the insertion sites on Chromosome I. The worm strain and transgenotype was named SHX3444 P*col-19*-COL-12::GFP (zjuSi337).

### 3.2. Scellseg Software Application for Single Worm Cutting

A series of images of different dosing groups was obtained by shooting and screening drugs according to the optimized conditions setting the relevant parameters, exporting the images directly from the Imagexpress PICO instrument, and then importing the images into Scellseg. After a series of images were subjected to machine learning, all the captured images were then imported into the software for further data processing to label worms after doing single worm cutting and exporting the single worm images.

### 3.3. Drug Screening and Data Analysis

COL-12 overexpressing worms cultured to adulthood were added to grenier 655,090 96-well plates, and a control group and a drug administration group were set up. The drug administration group added the prepared drug to the plates at a concentration of 100 μM. After ensuring the addition of the drug, the plates were placed in an incubator at 20 °C and incubated for 12 h. After 12 h, the drug plate was taken out from the incubator to shoot, the pictures were processed using Scellseg to get single worm pictures, and then further processed by imagej batch to get fluorescence data, and the mean gray value of the control group was set to 1.

### 3.4. Western Blotting

The worms were lysed with IP lysis buffer. The lysed proteins were then separated and transferred to PVDF membranes. Using anti-GFP, the anti-GFP (MBL, Japan) was diluted at 1:1000. The membrane was incubated at 4 °C overnight. The next day, membranes were incubated with anti-rabbit secondary antibody (Beyotime, Nantong, China) for 2 h at room temperature. The expression of GAPDH (Beyotime, Nantong, China) was used as the loading control. Images of blots were obtained by using Chemidoc Imaging System (Bio-Rad; Hercules, CA, USA).

### 3.5. RT-PCR

The worm RNA was extracted with easy RNA kit (Easy-Do, Beijing, China), HiScript II Q RT SuperMix for qPCR kit (Vazyme, Nanjing, China) was reverse transcribed to obtain cDNA, and finally, qPCR was performed with a qPCR fluorescence quantification kit (Vazyme, Nanjing, China) to obtain CT values for analysis.

### 3.6. Quantification and Statistical Analysis

All the images were analyzed with ImageJ (Bethesda, MA, USA). All statistical analyses used GraphPad Prism 7 (San Diego, CA, USA). The y-axis error bars for bar charts plotted from the mean value of the data was the Standard Error of the Mean (SEM). Two-way comparisons used unpaired t-test. Data were considered statistically different at *p* < 0.05. All experiments were repeated.

## 4. Conclusions

In conclusion, Danshensu, Lawsone and Sanguinarine are expected to be developed into drugs that affect collagen secretion and need further study to find out whether they are candidates of COL-12-related drugs. In future, we will verify the mechanisms and principles of action of compounds Danshensu, Lawsone and Sanguinarine through animal and clinical experiments.

## Figures and Tables

**Figure 1 molecules-27-08361-f001:**
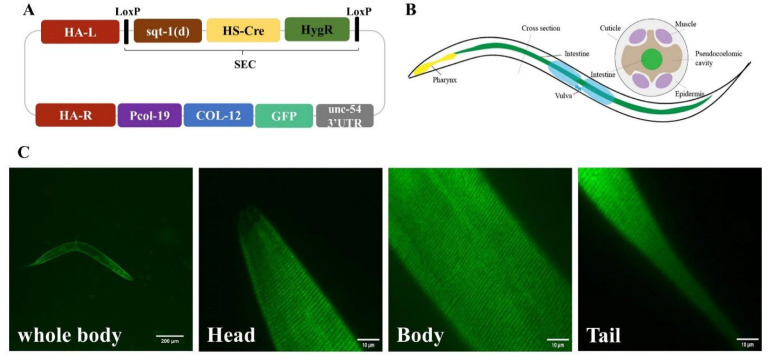
Construction of nematode models. (**A**) Restoration of templates; (**B**) Diagram of a nematode model; (**C**) single copy insertion transgenic nematode P*col-19*-COL -12::GFP model parts (Scale bars, 10 μm).

**Figure 2 molecules-27-08361-f002:**
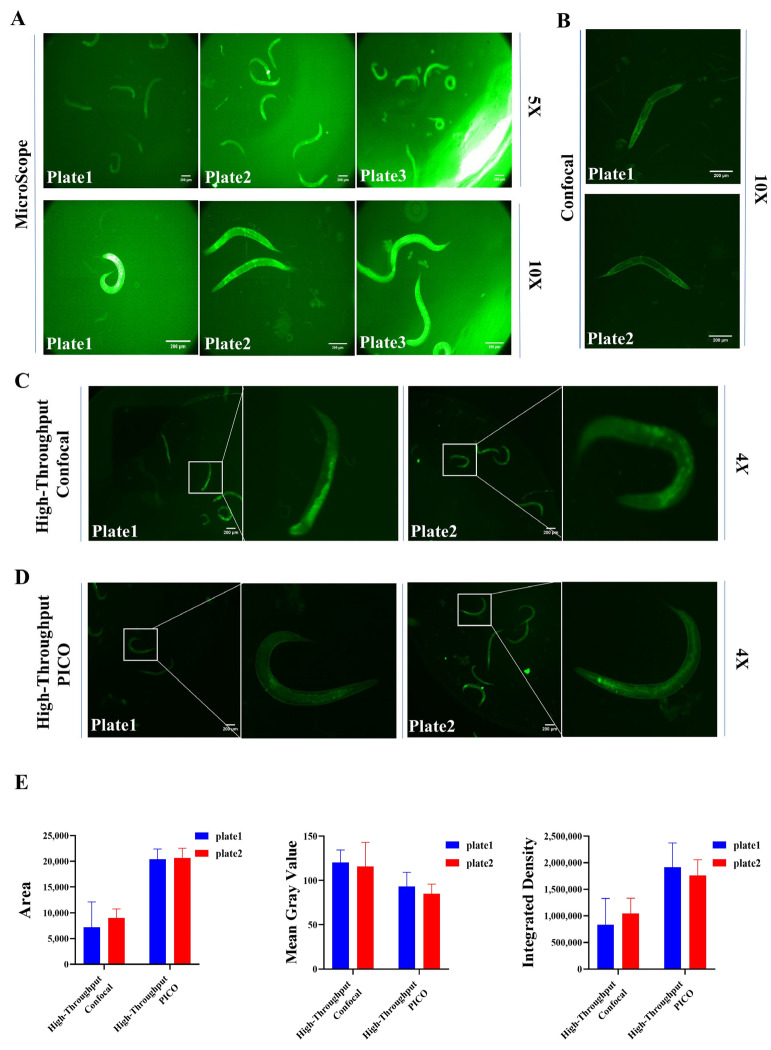
Optimization of imaging conditions (**A**) Image sampling with Microscope at different plate types and magnifications (Scale bars, 10 μm); (**B**) Image sampling with Confocal at different plate types and magnifications (Scale bars, 10 μm); (**C**) Image sampling with High-Throughput Confocal at different plate types and magnifications (Scale bars, 10 μm); (**D**) Image sampling with High-Throughput PICO at different plate types and magnifications (Scale bars, 10 μm); (**E**) Analysis of single nematode related data, *n* = 3.

**Figure 3 molecules-27-08361-f003:**
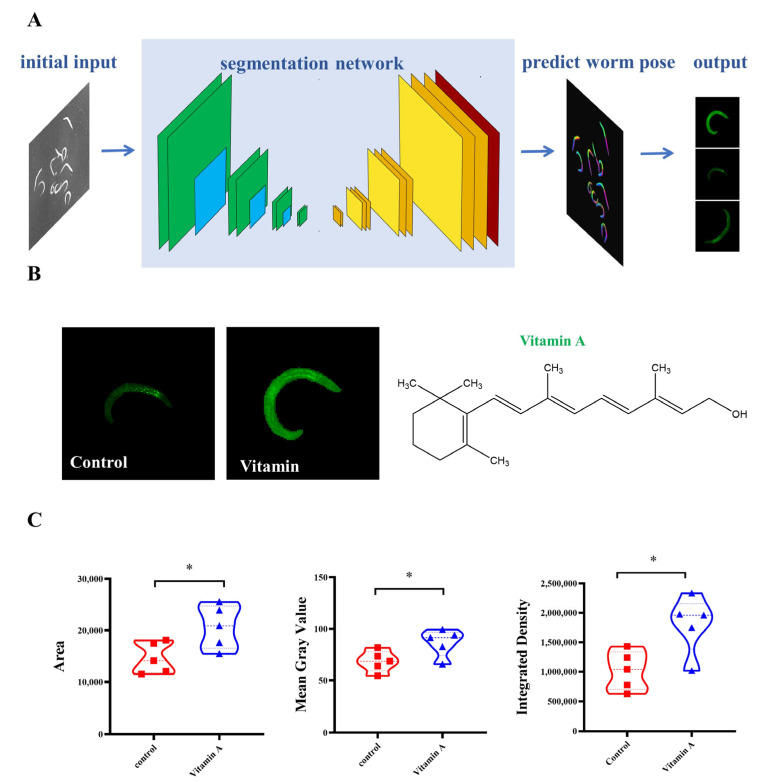
Single nematode cutting method for Scellsegs software (**A**) Software for single nematode isolation process; (**B**) Single nematode images obtained from the Control and vitamin A groups; (**C**) Analysis of single nematode related data for Control and Vitamin A groups (* *p* < 0.05 for the designated treatment vs. Control), *n* = 5.

**Figure 4 molecules-27-08361-f004:**
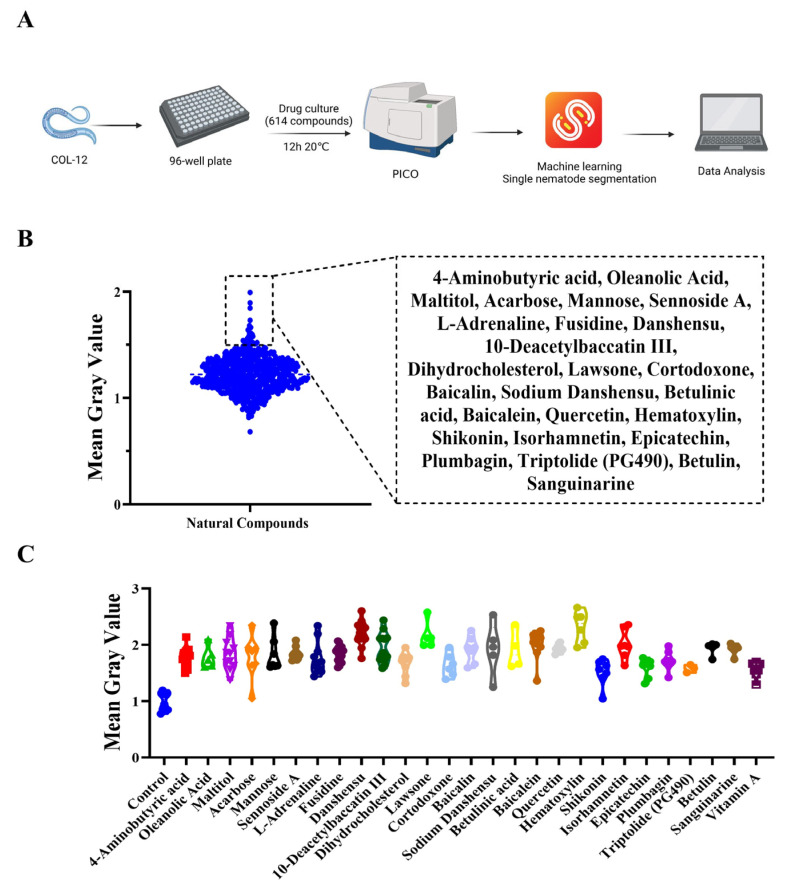
Compound library screening for promoting COL-12 expression (**A**) Drug screening flow chart; (**B**) Results of preliminary drug screening, where 614 compounds from the compound library were tested on the COL-12 model at a single concentration of 100 µM, *n* = 30; (**C**) Single nematode data for effective compound groups, *n ≈* 7.

**Figure 5 molecules-27-08361-f005:**
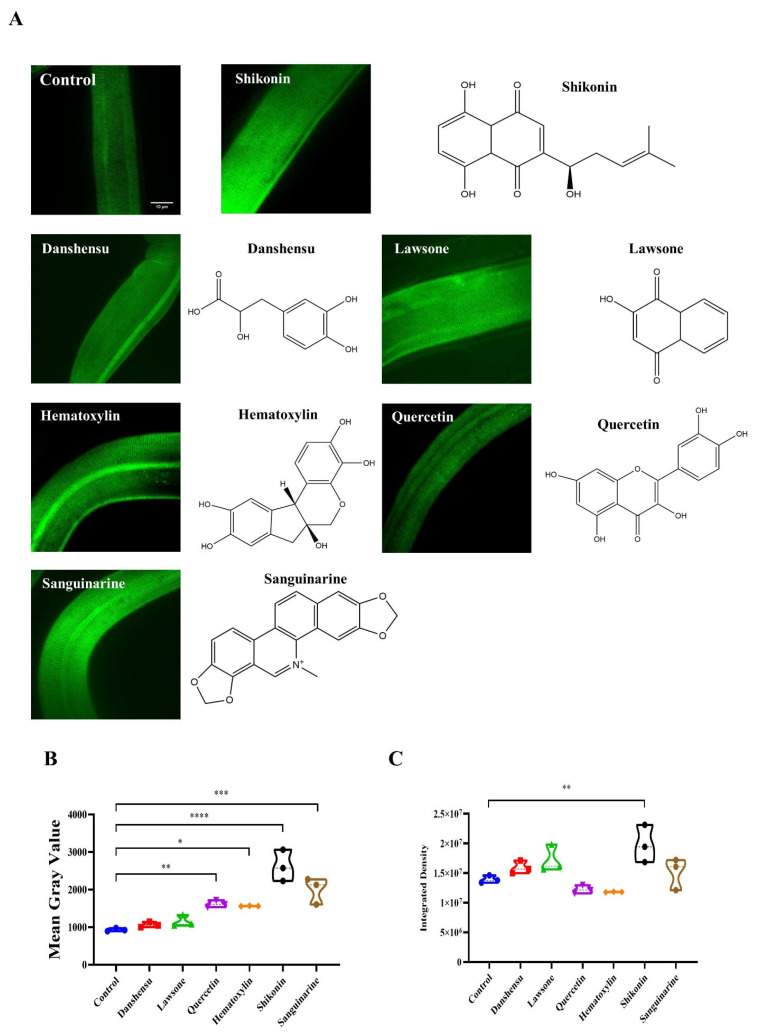
Fluorescence intensity analysis of compounds promoting COL-12 expression (**A**) COL-12 expression after co-incubation of different compounds imaged under 100× magnification (Scale bars, 10 μm); (**B**) Mean Gray Value of compounds promoting COL-12 expression, *n* = 3; (**C**) Integrated Density of compounds promoting COL-12 expression (* *p* < 0.05, ** *p* < 0.01, *** *p* < 0.001, **** *p* < 0.0001 for the designated treatment vs. Control), *n* = 3.

**Figure 6 molecules-27-08361-f006:**
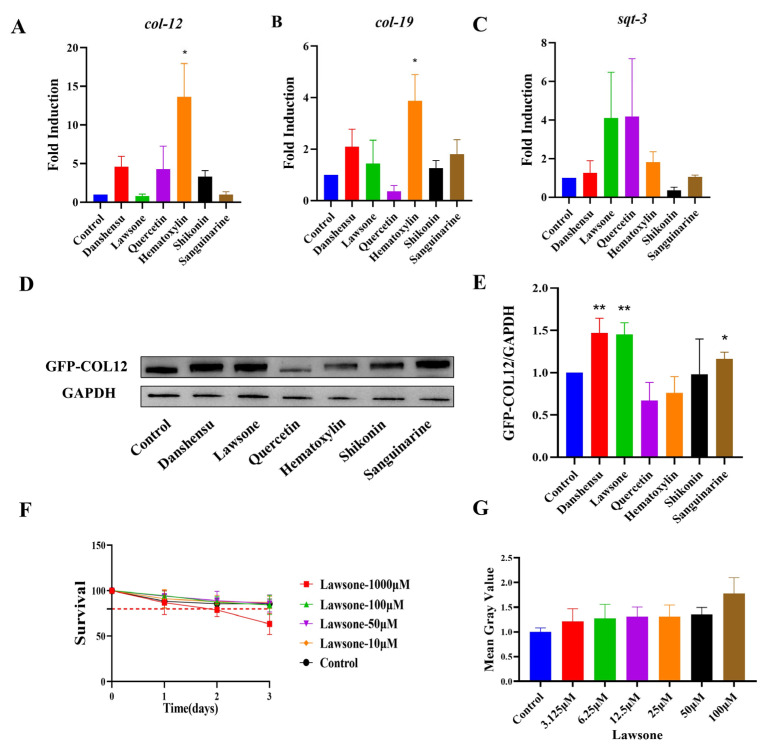
Quantitative analysis of compound-promoted COL-12 expression (**A**–**C**) Effects of different compounds on transcript levels of different collagen encoded genes (* *p* < 0.05 for the designated treatment vs. Control), *n* = 3; (**D**,**E**) Effects of different compounds on COL-12 expression level (* *p* < 0.05, ** *p* < 0.01 t test for the designated treatment vs. Control), *n* = 3; (**F**) Nematode survival rate, *n* = 3; (**G**) The quantitative-effect relationship of Lawsone, *n* = 3.

## Data Availability

The data used to support the findings of this study are available on request from the corresponding author.

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
