# Peer review of "Discovery of Natural Small Molecules Promoting Collagen Secretion by High-Throughput Screening in Caenorhabditis elegans"

_molecules, 2022, doi:10.3390/molecules27238361_

Round 1
Reviewer 1 Report
The manuscript by Wang Y and colleagues reports a chemical genetic screen using C elegans as a model. The authors screened >600 natural products to identify chemicals that can increase collagen 12 production. The screen identified a dozen candidates, some of which appear to regulate collagen biogenesis at post-transcriptional levels.
The study reports some useful information, but the reports read very descriptive; no molecular insight was provided regarding the potential mechanisms by which these chemicals regulate collagen biogenesis. I hope that the following comments can help the authors improve this manuscript.
-
Line 150, the data do not support the statement that these drugs affect Col-12 expression. They may regulate expression, secretion or turnover.
-
To make it easy to compare experiment to experiment, they should use a consistent y axis label in collagen quantification. Figure 4B, C uses very different scales.
-
They screens the compounds at single dose of 100uM, which is understandable given the large number of compounds involved. However, for validation, they should do a dose titration to determine the potency of these compounds, at least for the top hits.
-
Line 176-180, the authors claim that they checked the effect of the compounds on mRNA expression of several collagens, and no difference was observed. However, judged from Figure 6A-C, there are clearly some upregulations by drugs such as Danshensu and Hematoxylin etc.
-
Along the same line, when they claimed that the Col12 protein level was upregulated by drugs such as Danshensu, the protein level increase, as determined by immunoblotting, is only marginal. This is very different from the imaging data. Please clarify the discrepancy.
-
Although addressing the underlying mechanism may be beyond the scope of the current study, they should at least test another collagen with some top hits to see if the effect is specific to Col12.
-
Figure legends should include n value when statistical analyses were included.
Reviewer 2 Report
The authors describe a high throughput screening in a transgenic C.elegans model. The idea to develop an in vivo model system to identify compounds that might modulate collagen synthesis and secretion is interesting. However, the analysis presented here remains very superficial and the authors to do convincigly show that secretion is really improved after treament with specific compounds. I can agree the expression is increased (though only marginally) but from the title I would have expected that a impaired collagen secretion (as in several collagen-related pathologies) could be improved.
On what basis was COl12 selected? I understand that there is homology with COL6A5 in humans. Mutations in collagen VI mainly cause myopathies in humans which is not mentioned in the introduction. Also, not all triple helices are composed of different chains encoded by different genes! In humans there are only 28 collagens.
What is meant by aggregation forms (line 166)?
Looking at Figure 6A to C, I do not understand that none of the compounds affects the transcriptional level. Hematoxylin leads to a strong transcriptional induction but to a reduction at the protein level? I am not convinced by the quantitative anaylsis of the Western Blot.
I do not understand what is shown in Figure 6G....dose response relationship?
Conclusions: which 13 compounds were screened?
Several references are not properly displayed in the text.
English language should be revised by a native speaker or professional editing service.
Reviewer 3 Report
The submitted manuscript described a very sound model for testing drugs which can modulate the expression of Col 12 & Col 19.
I have a major comment: selection of the concentration of drugs to be tested at 100 µm.
What is the rationale for this dose? For some compounds, non-classical shape like bell shape for dose response may be observed.
As a second comment, for the compounds showing an effect on the expression of col 12 or col 19, it would have been good to explore other doses to establish a dose / response relationship.
Among the tested compounds, were some of them inhibitory on col 12 and/or col 19 expression.
As a third comment, it would be better for the quality of the manuscript to indicate the results with a couple of compounds without effects to be compared with the control response. This would help to compare a compound with a positive impact vs a compound with no visible impact.
As a last comment, when looking at the results, a single compound with real potential appears, shikonin. For the conclusion, it should be improved to reflect what should be really observed on col 12 or col 19 from a metabolic / clinical perspective. And try to elaborate possible mechanism of actions from what it is known elsewhere for this compound.
Round 2
Reviewer 1 Report
The authors have addressed all my comments.
Author Response
请参阅附件。
